# Design and Experimental Study of a Stepping Piezoelectric Actuator with Large Stroke and High Speed

**DOI:** 10.3390/mi14020267

**Published:** 2023-01-20

**Authors:** Qirui Duan, Yajun Zheng, Jun Jin, Ningdong Hu, Zenglei Zhang, Hongping Hu

**Affiliations:** 1Department of Mechanics, School of Aerospace Engineering, Huazhong University of Science and Technology, Wuhan 430074, China; 2Hubei Key Laboratory for Engineering Structural Analysis and Safety Assessment, Huazhong University of Science and Technology, Wuhan 430074, China; 3Shanghai Ruisheng Kaitai Acoustic Science and Technology Co., Ltd., Shanghai 201100, China; 4Wuhan Second Ship Design and Research Institute, Wuhan 430064, China

**Keywords:** piezoelectric stack, large stroke, high speed, finite element method, actuator

## Abstract

A stepping piezoelectric actuator is proposed with large stroke and high speed. The piezoelectric actuator consists of two symmetrical stators and a mover. The actuator can operate with a “double-drive, four-clamp” mode. The proposed actuator solves the problems of short stroke, low speed, and small load inherent in the currently published stepping piezoelectric actuators. By combining Workbench software with APDL language, finite element simulation and statics and dynamics analysis are carried out to guide the design of the actuator. The new piezoelectric simulation method can solve the difficulties regarding parameter setting and loading voltage on multiple interfaces for a complex piezoelectric model. Therefore, the novel method is helpful to develop the simulation of multilayer thin piezoelectric devices. The prototype of the actuator is developed and tested. Experimental results show that the actuator can run stably in the range of 0 to 600 Hz. The driving stroke is greater than 85 mm, the resolution can reach 535 nm, the maximum driving speed is 6.11 mm/s, and the maximum load is 49 N.

## 1. Introduction

With the continuous development of science and technology, high-precision and ultra-precision motion platforms have become research hotspots [1,2,3,4]. Driving systems with high precision, large stroke and high reliability are required in many fields, such as mechanical processing [5], microelectronics [6], precision optics [7], modern medicine [8], intelligent robotics [9], and aerospace [10]. The traditional precision driving mechanism mainly adopts mechanical transmission, using gears, levers, cams, sliders, precision ball screw pairs, and precision spiral wedges. Generally, these transmission systems have a complex structure and clearance. Therefore, the accuracy of the motion, positioning, and frequency response cannot meet the requirements of modern industrial development. To overcome these problems, many new precision actuators have been proposed, such as the magnetostrictive actuator [11], the shape memory alloy [12], the electrostrictive actuator, [13] and the piezoelectric actuator [14]. Among these actuators, the piezoelectric actuator has its own unique advantages, such as small size, good linearity, convenient control, no electromagnetic interference, high displacement resolution, good frequency response, low energy consumption, and low noise [15,16,17,18,19,20,21]. As a type of piezoelectric actuator, the stepping piezoelectric actuator can accumulate small displacement in a single step; the output stroke is infinite, theoretically. Furthermore, due to the small displacement in one step, both the large working stroke and high positioning resolution could be achieved by employing a dual-servo control strategy, which can greatly enhance the application potential of piezoelectric actuators [22]. Thus, the stepping piezoelectric actuator is very suitable for the application of precision positioning and high-precision mechanical transmission equipment, among others.

In recent years, based on piezoelectric elements, researchers have developed a variety of driving and positioning platforms, which are applied to biological cell micromanipulation, atomic manipulation, micro/nano indentation, aerial photography, and other systems [23,24,25,26,27]. However, the small output stroke—generally several microns—of piezoelectric elements limits the application of piezoelectric driving technology in many fields [28,29,30]. It is required that the actuators have the characteristics of large stroke, high speed, light weight, miniaturization, low cost, and low energy consumption [31,32,33]. The existing actuators in the industrial market cannot simultaneously meet the above requirements. Therefore, it is urgent to develop new piezoelectric actuators.

In this study, a stepping piezoelectric actuator is proposed, with large stroke and high speed, based on the principle of piezoelectric driving. Micro/nano output precision and large output stroke are combined to achieve a high precision linear output with large stroke, exhibiting broad application prospects in aerospace, mechanical processing, intelligent robots, and other related fields.

## 2. Working Principle and Structure Design of the Actuator

### 2.1. Working Principle of the Actuator

The stepping piezoelectric actuator is an important type of piezoelectric actuators. Its discovery was inspired by the inchworm’s crawling. The piezoelectric stacks are the power source of the stepping piezoelectric actuator. Multi-step macro displacements can then be realized by accumulating small displacements in a single step. Therefore, this type of actuator is characterized by high resolution, large stroke, and large load capacity [34,35,36,37]. Generally, a typical stepping piezoelectric actuator consists of three parts, with two clamping mechanisms on both sides and a driving mechanism in the middle. The working mode of this actuator is “clamp-drive-clamp”. The behavior is similar to the crawling motion of the inchworm. Thus, the actuator is also called an inchworm actuator.

The operating principle of the actuator is demonstrated in Figure 1a [38], in which 1 is the frame, 2 is the driving piezoelectric stack, 3 is the clamping piezoelectric stack, and 4 is the driving rod. Assuming the driving direction is to the right, the driving process can be divided into four steps. Firstly, on the right side, the upper and lower clamping piezoelectric stacks 13 and 23 are powered on and extended to clamp the driving rod. While on the left side, the upper and lower clamping piezoelectric stacks 11 and 21 are powered off and disengaged from the driving rod. Secondly, the driving piezoelectric stacks 12 and 22 are powered on and extended to drive the upper and lower clamping piezoelectric stacks 13 and 23, and the driving rod to the right. Thirdly, on the right side, the upper and lower clamping piezoelectric stacks 13 and 23 are powered off and disengaged from the driving rod. While on the left side, the upper and lower clamping piezoelectric stacks 11 and 21 are powered on and extended to clamp the driving rod. Finally, the driving piezoelectric stacks 12 and 22 retract as the voltage decreases linearly, and the upper and lower clamping piezoelectric stacks 11 and 21, along with the driving rod, are driven to the right. If the driving rod is to be moved in reverse, the timing of the clamping piezoelectric stacks on the left and right sides should be swapped.

The movement of the driving rod requires different piezoelectric stacks to produce telescopic movement in different time steps, and the motion between different piezoelectric stacks needs to be coordinated. The signal generator can generate the required triangle wave and square wave to control the joint coordinated movement between piezoelectric stacks, as shown in Figure 1b.

### 2.2. Structure Design of the Actuator

The overall structure of the actuator is shown in Figure 2a. The actuator consists of two stators, a mover, i.e., a rolling rail, two adjusting platforms, and a baseplate to support all the parts. The stators are driven and clamped by piezoelectric stacks, and the linear motion of the mover is realized by the stepping principle, as introduced in Section 2.1.

The stator consists of two clamping mechanisms on both sides and a driving mechanism in the middle. Two symmetrical stators enable the actuator to be expanded to a “double-drive, four-clamp” mode. Compared to the previous “single-drive, double-clamp” mode, the stability and load capacity of the actuator has been greatly improved. As shown in Figure 2b, the driving mechanism is composed of a driving stack, two fixed holes, and four S-shaped flexure hinges. Its main function is to output displacement and thrust [39]. The clamping mechanism is composed of a clamping stack, a friction foot, and two S-shaped flexible hinges. Its main function is to make the output force of the clamping stack fully act on the surface of the mover, to make the clamping stable and reliable, and to ensure the normal force required for generating static friction.

In order to provide the stator with large output displacement and good stability, and to further ensure the reliability and accuracy of the mechanical system of the actuator, a large allowable stress *σ* and a medium Young’s modulus *E* of the material are required [40,41]. Therefore, 65 Mn spring steel is selected for the material of the stator. Additionally, the baseplate is made of aluminum, for cost reduction. The driving stack and clamping stacks are piezoelectric stacks pst 150/5 × 5/20 and 150/5 × 5/10 (Harbin Core Tomorrow Science & Technology Co., Ltd, Harbin, China), respectively. The number 150 means that the driving stack is operating at 0~150 V; 5 × 5 denotes that the cross section sizes are 5 mm × 5 mm. The last number, 20 or 10, indicates that the maximum output displacement is 20 μm or 10 μm, respectively. In addition, the maximum output forces are both 1600 N. For piezoelectric stacks, NCE51 is selected as the piezoelectric material, and Cu is used as the electrode material. Material parameters are shown in Table 1.

## 3. Simulation on the Actuator

To optimize the performance of the proposed piezoelectric actuator, the radius of the S-shaped flexure hinge is determined by the structural parameter analysis. The static analysis and dynamic analysis of the actuator are carried out to ensure the safety of the structure. The finite element software Workbench is used for the following simulation.

### 3.1. Structural Parameter Analysis of the Actuator Stator

The stator is an important component to transfer the output displacement and force of the piezoelectric stack. Therefore, the output performance of the actuator is fundamentally determined by the stator [42]. In addition, the stiffness of the S-shaped flexure hinge is the key parameter to determine the performance of the stator. Everything has two sides. On one side, the output displacement and force of the piezoelectric stack is reduced greatly by the large stiffness of the hinge. On the other side, due to the small stiffness of the hinge, the stator is easily damaged due to the large strain [43]. Furthermore, although the output displacement increases during the step linearly increasing the voltage of the piezoelectric stack, the output force of the piezoelectric stack decreases with the increase in the output displacement. As the installation sizes of the piezoelectric stacks are determined, only the radii *R*_31_ and *R*_32_ of the S-shaped flexure hinge need to be optimized, where *R*_31_ and *R*_32_ are the radii of the S-shaped hinges in the driving and clamping mechanisms, respectively. First, to optimize radius *R*_31_, the boundary conditions are set, as shown in Figure 3. The A surfaces of the two holes are fixed. The displacements of the stator bottoms in the *Y* and *Z* directions are constrained, but that in the *X* direction is free. According to the distortion energy strength theory, the von Mises stress can be used as the equivalent stress for the strength check.

The dependence of the maximum equivalent stress of the stator and the output displacement in the *X* direction of the friction foot upon the radius under different external loads is investigated. It can be seen from Figure 4 that when the force of 100 N acts on surface B, the maximum equivalent stress of the stator and the *X* direction displacement of the friction foot both decrease with the increase in radius *R*_31_. This is because the stiffness of the structure increases with the increase in radius *R*_31_. As the force remains unchanged, the deformation and strain decease, and the stress decreases accordingly, while when the displacement of 10 μm is applied on surface B, the maximum equivalent stress of the stator increases with the increase in radius *R*_31_. The phenomenon mainly stems from the fact that the displacement remains unchanged, as does the maximum deformation. However, with the increase in stiffness, the stress increases. It can be further seen from Figure 4a that the maximum equivalent stress is less than the allowable stress of the material, provided that the radius *R*_31_ is greater than 0.6 mm. For the driving stack pst 150/5 × 5/20, the maximum output displacement is 20 μm. Thus, one side of the clamping mechanism moves with a maximum distance of less than 10 μm. The radius *R*_31_ should be between 0.6 and 1.0 mm when considering maximum output displacement and the safety of the structure. The radius *R*_31_ of 1.0 mm is used as manufacturing size. In the simulation results, the displacement of the friction foot in the *X* direction was 10.2 μm. The maximum equivalent stress of the stator was 128 MPa, which is less than the allowable stress of 65 Mn spring steel 432 MPa.

The radius *R*_32_ of the S-shaped hinge in the clamping mechanism is further optimized. The force of 100 N is applied to the clamping mechanism in the finite element analysis. After optimization, *R*_32_ = 0.4 mm. The displacement of the friction foot in the *Y* direction is 10.5 μm. The maximum equivalent stress of the stator is 293 MPa, which is within the safety range.

### 3.2. Static Analysis of the Actuator

The performance of the actuator is further investigated by installing the piezoelectric stacks into the stator. Only one of the two stators is analyzed, since they are mounted symmetrically in the actuator. In software Workbench, the ACT plug-in is introduced for the piezoelectric simulation [44]. In addition, for multilayer piezoelectric stack, the polarization direction and voltage of different piezoelectric layers need to be set alternately. Because the thickness of each layer is very thin and the number of the piezoelectric layers is large, it is difficult to select faces and bodies in the Graphical User Interface (GUI) of Workbench. A new simulation method is proposed to solve the problem by the combination of Workbench software and the APDL language, which is abbreviated to Workbench + APDL. The simulation process of Workbench + APDL is shown in Figure 5. The establishment of the geometric model, the finite element model, and post-processing can be completed with programs written in APDL language. The finite element model includes assigning material properties to the geometric model, meshing, and boundary conditions.

First, the feasibility of the simulation method is verified. The pst 150/5 × 5/20 piezoelectric stack is calculated by ANSYS APDL and Workbench + APDL, respectively. The numerical results show that these two methods are in good agreement with each other, and the maximum output displacements are both 21.74 μm. The factory technical index of the piezoelectric stack is 20 ± 2 μm. The simulation error is 8.7%, less than 10% of the factory error. Thus, the Workbench + APDL method is applied in the following calculation. The proposed simulation method provides a new idea for the development of multilayer piezoelectric structures.

The boundary conditions at the outer boundaries are the same as those of the structural parameter analysis in Section 3.1. The output performance and structural safety of the stator are analyzed by applying 150 V DC voltage to the driving stack. As can be seen from Figure 6, the driving stack can drive the clamping mechanism to move freely along the *X* direction. The maximum displacement of the friction foot along the *X* direction is 10.8 μm. The maximum equivalent stress of 149 MPa, which is less than the allowable stress of 65 Mn spring steel, appears at the S-shaped flexure hinge. Therefore, the security of the actuator can be guaranteed.

### 3.3. Dynamic Analysis of the Actuator

#### 3.3.1. Modal Analysis of the Stator

The proposed actuator is designed to operate below 1 kHz. When the operating frequency approaches the natural frequency of the actuator, the resonance will occur with large amplitude, which will seriously affect the performance of the actuator. In the worst case scenario, the actuator can even be damaged. Therefore, the first six natural frequencies are calculated by the modal analysis of the stator; these are listed in Table 2. The first natural frequency of the stator is 3225 Hz, which is much larger than 1 kHz. Thus, the resonance can be avoided.

#### 3.3.2. Transient Dynamic Analysis of the Actuator

The transient dynamic simulation of the actuator is carried out below. The boundary conditions are shown in Figure 7a. The two mounting holes of the stator are fixed, and the lower surfaces of the stator and the mover restrict the displacement in *Y* and *Z* directions, while the *X* direction is free. The contact between the stator and mover is set as friction. The friction coefficient is set as 0.1, since the friction coefficient between steel and steel is between 0.1 and 0.15 [45]. The signal of the driving voltage applied to the piezoelectric stacks is given in Figure 1b. When the signal with the peak voltage of 150 V and the frequency of 10 Hz is applied, the maximum equivalent stress is 213 MPa, which is still less than the allowable stress of 65 Mn spring steel. Under such an operating condition, the displacement curve of the mover in one period is shown in Figure 7b. The operation of the actuator is stable, but it jumps backward and forward at the moments when the mover is being clamped by the stators. This is mainly due to the small bending stiffness of the S-shaped flexure hinge. Take the right driving signal as an example. When the square wave driving signal is applied to the clamping stack, on the one hand, the right friction foot moves in the positive *Y* direction. On the other hand, it moves in the negative *X* direction. This is because the clamping mechanism is bent, and the bending deflection on the friction foot is in the negative *X* direction. When the right friction foot clamps the mover, the displacement in the positive *Y* direction will generate the normal pressure required for friction between the stator and the mover, while the displacement in the negative *X* direction will cause the right friction foot to drive the mover to jump backward at the moment of the mover being clamped by the stator. Similarly, at the moment the left friction foot clamps the mover, the mover jumps forward. Over a period of time except these two moments, the mover gradually moves in the positive direction of the *X* axis. Finally, the displacement in one period can be as large as 20.8 μm.

## 4. Experiments

### 4.1. Experimental System

To measure the proposed actuator performance, an experimental platform is built, as shown in Figure 8. The voltage signal from the signal generator is amplified by the special power amplifier for the piezoelectric stack; then, the amplified signal acts on the piezoelectric stacks of the actuator. The signal generator is FY8300S (Zhengzhou FeelTech Co., Ltd, Zhengzhou, China). The driving stack and clamping stacks are piezoelectric stacks pst 150/5 × 5/20 and 150/5 × 5/10 (Harbin Core Tomorrow Science & Technology Co., Ltd, Harbin, China), respectively. The special power amplifier is RH61-A (Harbin Soluble Core Technology Co., Ltd, Harbin, China). The actuator is processed by laser wire cutting technology. The laser displacement sensor is LK-H050 (KEYENCE). The laser displacement sensor and computer are used to collect and store the data, respectively. The rolling guide rail is VRT1085A-25, produced by THK. The material of the rail is steel. The use of alumina ceramic plates can increase friction and enhance heat dissipation, improving wear resistance. In the experiment, we tested the bonding between the alumina ceramic plate and the metal guide rail. However, the alumina ceramic plate easily falls off the guide rail or friction foot, especially under heavy loading. Therefore, strengthening the bonding strength between the alumina and the metal remains a difficult problem. In future research, alumina will be used as the material for the whole friction foot, which can solve the problem of low bond strength and improve the friction coefficient and wear resistance.

To avoid surrounding vibration interference, the actuator and laser displacement sensor are placed on the air floating vibration isolation platform, while the other equipment is placed on another test table. The experiment is carried out at room temperature.

### 4.2. Actuator Speed

The actuator operates at alternating steps by using triangle wave and square wave signals shown in Figure 1b. The speed curves of the actuator, without load, are shown in Figure 9a. It can be noted that the speed of the actuator is dependent on the driving voltage and frequency. When the frequency is fixed, the speed increases with the increase in the voltage. On the other hand, when the driving voltage is fixed, the speed increases first, then decreases with the increase in frequency. This is because the output displacement of the piezoelectric stack will attenuate with the increase in frequency [46]. The larger the frequency, the more severe the attenuation. When the speed attenuation due to the increase in frequency is greater than the speed increase due to the increase in frequency, the actuator speed will decrease. The experimental results show that the actuator can operate stably at 0~600 Hz. When the driving voltage is 150 V with the frequency of 500 Hz, the maximum speed of the actuator, without load, can reach 6.11 mm/s, as shown in Figure 9b. However, the speed of experimental result is less than that of simulation result of 10.4 mm/s, introduced in 3.1.2. The difference can result from machining and assembly errors and the preloads of piezoelectric stacks.

### 4.3. Load Capacity

The load capacity is an important index to measure the performance of the actuator. Two types of load capacities are measured. The first one is to test the load of the actuator by placing a weight on the mover, and the second one is to test the thrust of the actuator by hanging a weight on the mover through a fixed pulley. A nylon string runs around a fixed pulley and connects the mover of the actuator and the weight, so that the mover can run in the horizontal track while the suspended weight rises vertically. Therefore, the second load type is also called the load against gravity.

In order to obtain the load capacity of the actuator with a driving voltage of 120 V and 400 Hz, the displacement versus time under different load is obtained. It should be pointed out that the load belongs to the first type. As shown in Figure 10a, the actuator can operate stably at a speed of 2.08~3.60 mm/s under different mass loads. Moreover, the dependence of the speed upon thrust is shown in Figure 10b. The speed of the actuator decreases rapidly with the increase in the second type of load. Specifically, when the load mass reaches 1 kg, the speed of the actuator decreases to 8 μm/s. Therefore, the maximum thrust of the actuator is 9.8 N.

### 4.4. Actuator Resolution

Actuator resolution refers to the step distance at which the actuator can operate stably under a minimum driving voltage. Resolution is an important index to measure the motion accuracy of the actuator. In the experiment, the frequency of the voltage is set to 1 Hz, and the resolution of the actuator is tested by adjusting the amplitudes of the driving voltage and clamping voltage. It was found from the test that when the clamping voltage is lower than 30 V, the actuator could not generate enough clamping force due to assembly error. When the clamping voltage is fixed at 30 V, the minimum driving voltage of the actuator is 12 V, since the actuator cannot operate stably until the driving voltage is increased to 12 V. In order to accurately obtain the resolution of the actuator, 4 cycles are measured in the experiment, as shown in Figure 11. From the figure, the mover of the actuator moves 2.14 μm in 4 cycles, so the actuator resolution is calculated as 535 nm. In contrast, the jumping backward motion of the mover is intensified at the clamping moment in the simulation results, as shown in Figure 7. This is mainly because the amplitude of the clamping voltage in the simulation is much larger than that in the experiment. Similarly, the mover jumps forward at the clamping moment of the left friction foot. Thus, the resolution of the actuator is greatly affected by the clamping voltage.

The mechanical output performance of the proposed actuator is compared with the same type of stepping piezoelectric actuators, as listed in Table 3. It can be seen that the proposed actuator exhibits obvious superiority in speed, stroke, and load. These experimental results verify that the proposed actuator can achieve nanometer resolution, long range motion, high speed, and large load. It will have broad application prospects in aerospace, mechanical processing, intelligent robots, and other fields.

## 5. Conclusions

In summary, a novel piezoelectric actuator, with large stroke and high speed, was proposed. This actuator has overcome the disadvantages of small stroke and slow speed. The present work has broad application prospects in aerospace, mechanical processing, intelligent robots, and other fields. The conclusions can be drawn as follows:

(1) Two symmetrical stators were designed for the actuator so that the actuator can operate with a “double-drive, four-clamp” mode to provide better operating stability and larger load capacity.

(2) A new piezoelectric simulation method of Workbench + APDL was proposed, providing the convenience of setting parameters and boundary conditions for the multilayer ultra-thin piezoelectric structure, and presenting a new idea for the development of simulations on multilayer piezoelectric devices.

(3) The experimental study was carried out and the results show that the actuator can run stably in 0~600 Hz, the driving stroke can reach 85 mm, the resolution can reach 535 nm, the maximum driving speed is 6.11 mm/s, and the maximum load is 49 N. Compared with the currently published types of stepping piezoelectric actuators, the proposed actuator exhibits obvious superiority regarding speed, stroke, and load.

## Figures and Tables

**Figure 1 micromachines-14-00267-f001:**
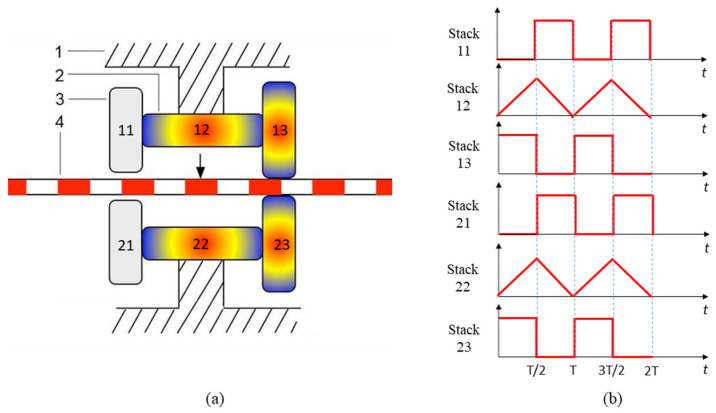
Diagrams of the stepping piezoelectric actuator: (**a**) operating principle; (**b**) sequence of the driving voltage.

**Figure 2 micromachines-14-00267-f002:**
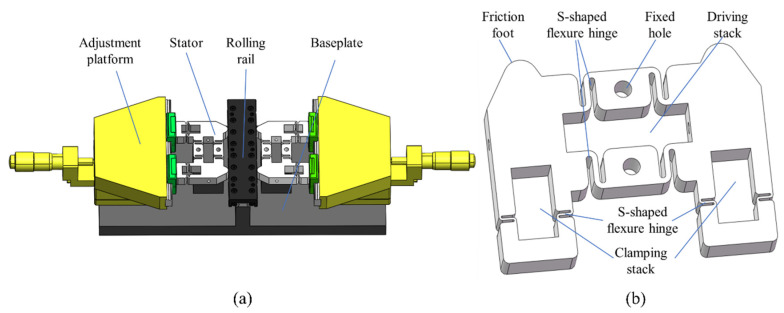
The stepping piezoelectric actuator: (**a**) assembly of construction; (**b**) stator.

**Figure 3 micromachines-14-00267-f003:**
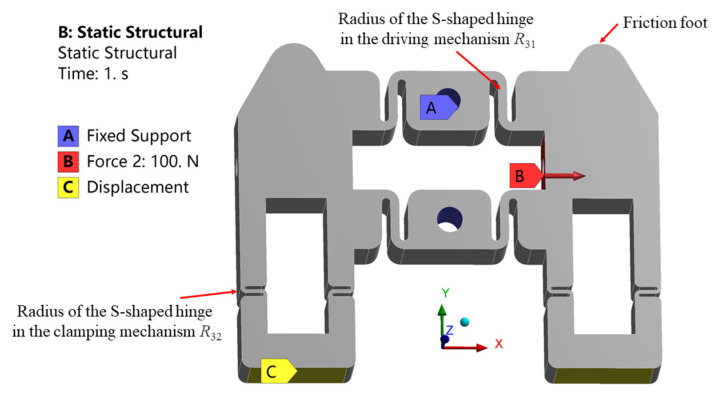
Boundary conditions for simulation of the stator.

**Figure 4 micromachines-14-00267-f004:**
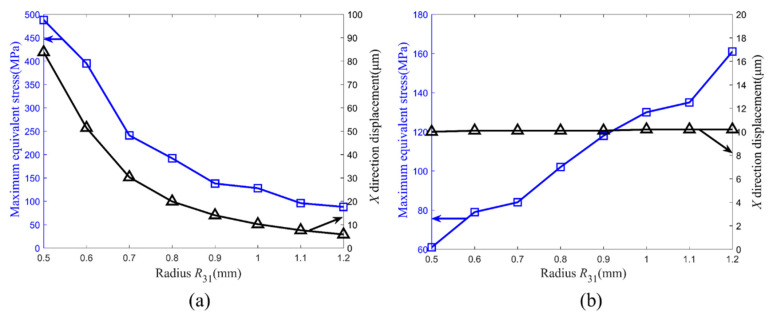
Maximum equivalent stress and output displacement versus radius *R*_31_ under different external loads: (**a**) force of 100 N; (**b**) displacement of 10 μm.

**Figure 5 micromachines-14-00267-f005:**
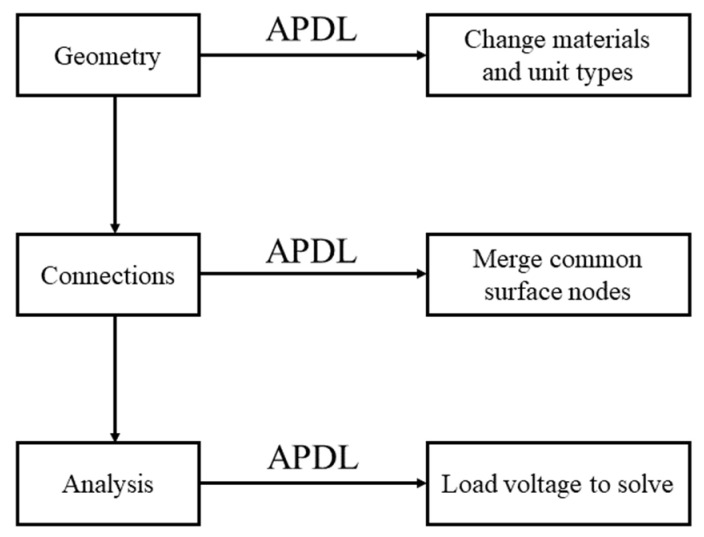
Process diagram of the simulation of Workbench + APDL.

**Figure 6 micromachines-14-00267-f006:**
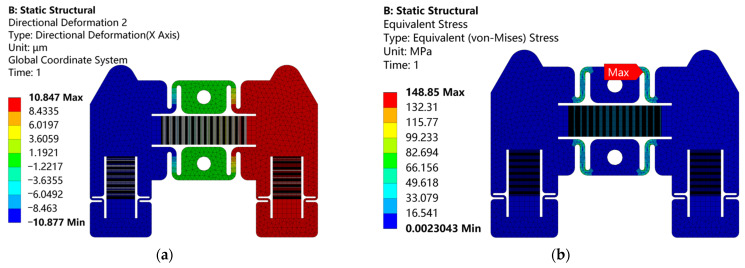
Nephogram of the actuator in static simulation: (**a**) displacement; (**b**) stress.

**Figure 7 micromachines-14-00267-f007:**
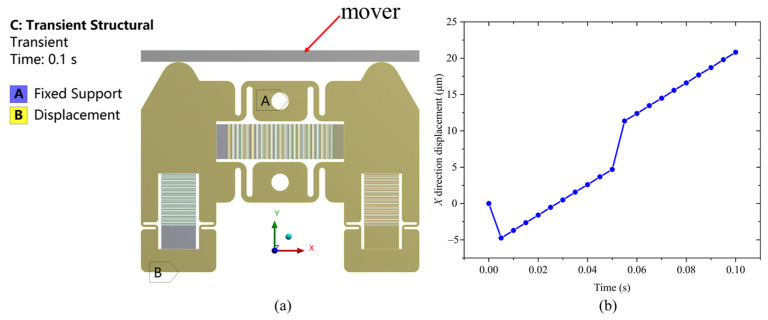
(**a**) Boundary conditions of the actuator in transient dynamic simulation; (**b**) displacement curve of the mover along the *X* direction in one period, where the driving and clamping voltage amplitudes are 150 V and 120 V, respectively.

**Figure 8 micromachines-14-00267-f008:**
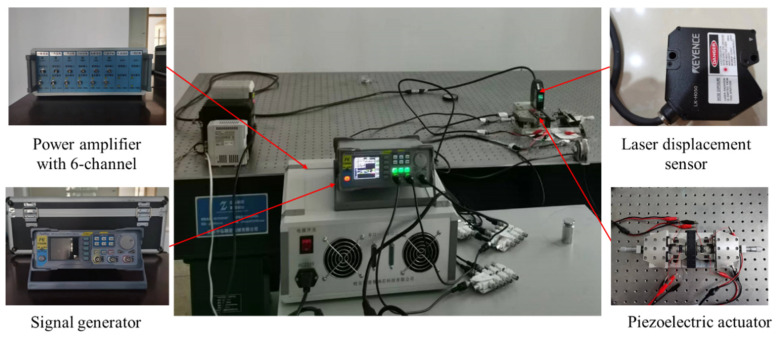
Experimental platform of the piezoelectric actuator.

**Figure 9 micromachines-14-00267-f009:**
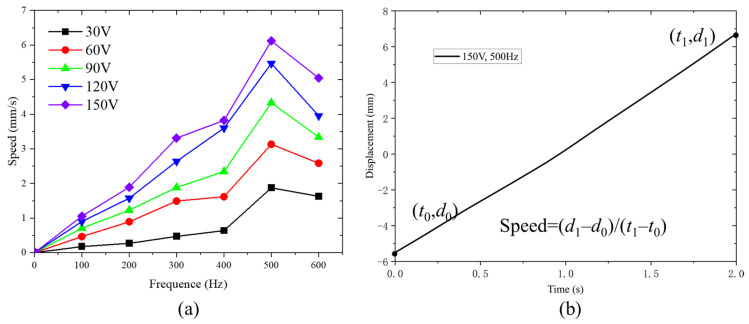
Speed test results of the actuator: (**a**) speed versus frequency under different driving voltage; (**b**) time history of the displacement of driving voltage with 150 V and 500 Hz.

**Figure 10 micromachines-14-00267-f010:**
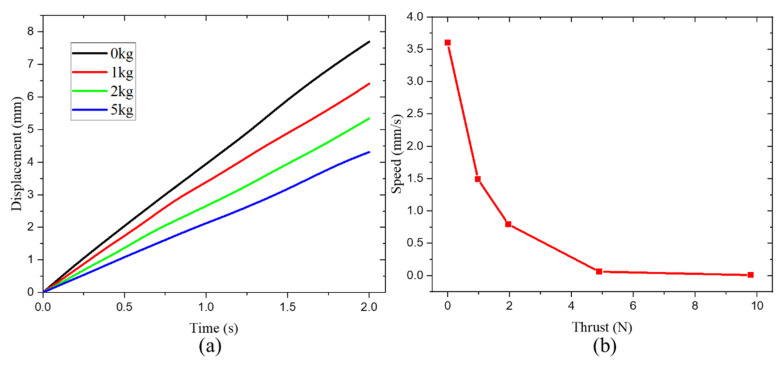
Test results of driving voltage of 120 V and 400 Hz: (**a**) time history of displacement under different load; (**b**) speed versus thrust.

**Figure 11 micromachines-14-00267-f011:**
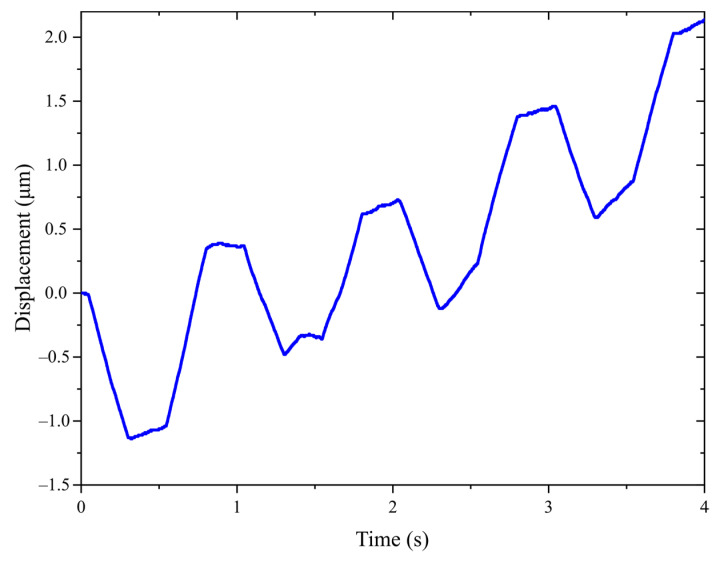
The step distance test curve under driving and clamping voltage amplitudes are 12 V and 30 V, respectively.

**Table 1 micromachines-14-00267-t001:** The material parameters of the piezoelectric stacks.

Materials	Density *ρ* (kg/m^3^)	Young’s Modulus *E* (GPa)	Poisson’s Ratio *μ*	Charge Constants (10^−12^ C/N)		
				*d* _31_	*d* _33_	*d* _15_
Piezoelectric material (NCE51)	7850	\	\	−208	443	669
Electrode material (Cu)	8600	110	0.34	\	\	\

**Table 2 micromachines-14-00267-t002:** The first six natural frequencies of the stator.

Modal Order	1st	2nd	3rd	4th	5th	6th
Natural frequencies (Hz)	3225	3256	9325	9430	16,804	16,820

**Table 3 micromachines-14-00267-t003:** Performance comparison of stepping piezoelectric actuators.

Authors	Speed (mm/s)	Stroke (mm)	Load (N)	Thrust (N)	Size (mm)	Number of Stacks
Xue et al. (2016) [30]	0.45	N/A	N/A	15	135 × 120 × 35	4
Chen et al. (2019) [47]	0.063	70	N/A	1.6	N/A	2
Zhang et al. (2020) [40]	4.76	45	13	N/A	108 × 75 × 48	2
Deng et al. (2022) [41]	0.16	40	N/A	12.3	Φ34 × 40	3
This work	6.11	85	49	9.8	208 × 85 × 48	6

## Data Availability

Data underlying the results presented in this paper are not publicly available at this time, but may be obtained from the authors upon reasonable request.

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
