# Peer review of "Design and Experimental Study of a Stepping Piezoelectric Actuator with Large Stroke and High Speed"

_micromachines, 2023, doi:10.3390/mi14020267_

Round 1

Reviewer 1 Report

In this paper, a stepping piezoelectric actuator with two symmetrical stators and a mover is proposed to with large stroke and high speed. Although the prototype is developed with experimental tests, there are some issues that should be explained in detail.

 The specific comments are as follows:

(1)  In the introduction, the references is not adequate. More state of the art works about inchworm type actuators should be cited, analysis and summarized. Please refer to [1]  [2] [3] [4].

[1] Wang, L., Chen, W., Liu, J., Deng, J., & Liu, Y. (2019). A review of recent studies on non-resonant piezoelectric actuators. *Mechanical Systems and Signal Processing*, *133*, 106254.

[2] Li, J., Huang, H., & Morita, T. (2019). Stepping piezoelectric actuators with large working stroke for nano-positioning systems: A review. *Sensors and Actuators A: Physical*, *292*, 39-51.

[3] Ling, J., Chen, L., Feng, Z., & Zhu, Y. (2022). Development and test of a high speed pusher-type inchworm piezoelectric actuator with asymmetric driving and clamping configuration. *Mechanism and Machine Theory*, *176*, 104997.

[4] Ma, X., Liu, Y., Deng, J., Gao, X., & Cheng, J. (2023). A compact inchworm piezoelectric actuator with high speed: Design, modeling, and experimental evaluation. *Mechanical Systems and Signal Processing*, *184*, 109704.

(2) The contributions and novelty of this work is not clear. 

(3) How to get the conclusion that the actuator resolution is 535 nm from Fig. 11. 

(4) Please add the comparison of the size in Table 2. 

(5) The detailed information of the experimental system should be provided. 

Reviewer 2 Report

Dear Authors,

My review report is in the attachment. 

Kind Regards

Round 2

Reviewer 1 Report

The comments have been addressed in the last round. 

Reviewer 2 Report

Dear Authors,

You have revised the manuscript according to my remarks. I have no further questions and comments.

The manuscript can be published after minor revision (format mistakes, methodological errors etc.)

Kind Regards